# Conflicting Maps: How Legal Perspectives Could Minimize Zoning Cancellation in Republic of Korea

**Jung-kyun Moon [1], Seon-bong Yoo [2],\*, Hong-gyoo Sohn [3] and Yonng-sun Cho [4]**

1   Legal Policy Research Institute, 1, Uisadang-daero, Yeongdeungpo-gu, Seoul 07233, Korea; sepperse@naver.com
2   Division of Law and the Graduate School of Construction Legal Affairs, Kwangwoon University, 20 Kwangwoon-ro, Nowon-gu, Seoul 01897, Korea
3   School of Civil & Environmental Engineering, A463, Engineering Hall 1, Yonsei University 50 Yonsei-ro, Seodaemun-gu, Seoul 03722, Korea; sohn1@yonsei.ac.kr
4   Hanmac Engineering, 301-ho Dream Plaza 2, 6-8 Sinseonghwaro 46 Beon-gil, Seowon-gu, Chungcheongbuk-do, Cheongju 28164, Korea; hanmac65@hanmail.net
\*   Correspondence: sbyu@kw.ac.kr; Tel.: +82-2-940-5401

**Abstract:** The purpose of this paper is to propose legal and policy enhancements that may prevent the cancellation of the legal force of zoning due to discord with the Korean Land Use Regulation Map (LURM) and secure legal stability. The legal force of zoning has been canceled because of the discordance of the LURM with past cadastral maps, and this has led to confusion regarding zoning decisions and even the postponement and cancellation of public projects. Here, the causes of LURM discordance and legal cancellation of zoning were identified and evaluated through judicial precedents. We found that improper use of data and adoption of tolerance caused the cancellations. To remedy these problems, we suggest the disclosure and application of cadastral computerized data instead of serial cadastral maps during LURM production activities to justify the legal adoption of allowable errors. We also recommend the widespread introduction of legal fiction for the rapid production of digital cadastral maps. Zoning cancellation could be minimized through such enhancements, and the map could allow people to visualize elements more conveniently. Moreover, this study aims to expand relevant legal mapping.

**Keywords:** land use regulation map; zoning; cadastral computerized data; digital cadastral map; legal fiction; allowable error; legal mapping

## 1. Introduction

Domestic and foreign researchers have attempted to identify the reciprocal relationships and influences of zoning and zoning maps, which are used to visualize a given area with respect to legal and functional perspectives [1–4]. The importance of zoning maps has been emphasized in historical visualizations, and prior research regarding urban planning and development has asserted that zoning maps are legal documents rather than simple maps. It has been further suggested that issues such as discordances due to an overlap of historic and current zoning maps must be alleviated during the preparation process, potentially through technological enhancements [1–4].

This study extends such analysis work and examines the legal characteristics and problems that have been encountered during the implementation of zoning maps through the Korean Land Use Regulation Map (LURM), a type of zoning map in Korea used to construct proposals from legal perspectives.

Moreover, the authors of this work believe that such studies of LURM cases in Korea have broad applicability because each nation has prepared zoning maps by utilizing topographic maps and cadastral maps. With the development of computer technology and digital maps, many zoning maps are expected to evolve [1–4]. Therefore, the possible

enhancements suggested in this research regarding the legal cancellation of zoning due to discordances with cadastral maps in the past—highlighted as the LURM problem—could be useful as a guiding framework and reference to other nations.

In 2005, the Framework Act on the Regulation of Land Use (FARLU) was established in Korea to protect the people's property rights from abuse by zoning regulations and to ensure transparency in administrative processes, even in cases of public use. In addition, zoning has the legal effect of limiting individuals' property rights (right of use) only after producing the LURM, a requirement unique to the FARLU, and registration in the official gazette and Land Use Regulation Information System (LURIS) is required. The time permitted until LURM production and notification is a maximum of two years [5]. The LURM is defined as a map conforming to legal effects through illustration of the zoning and scope of land use right restrictions. However, due to discordances between the LURM and cadastral maps in terms of boundaries and areas, the legal effect of zoning is being canceled.

Specifically, local governments have canceled approximately 1400 cases of zoning during the past three years because of LURM discordances [6]. Zoning has become a confusing issue, with public projects being suspended or postponed. Furthermore, local governments are spending more than 40 billion won (approximately USD $23 million) each year for additional mapping [6]. The area of the current LURM is roughly 4.3 times (432,390 km$^2$) the territory of the Republic of Korea (99,720 km$^2$) This could be owing to overlapping zonings on the same land mass [6].

Many studies have considered technical and engineering approaches to improve the conformity of maps, such as in LURM discordance cases, including data integration, conflation, and computer-based systems. Conflation, which corrects discordances, involves the production of a new map that can be used through one-to-one matching, integration, or overlay [7–9]. However, to the best of the authors' knowledge, no studies have sought to maintain the legal effects of maps through improvements based on legal principles or minimize or prevent the cancellation of zoning.

This paper intends to clarify why zoning is being canceled due to the LURM by reviewing the related local and foreign documents, with case studies and Supreme Court precedents before and after the introduction of the LURM, to minimize future cancellations of zoning and propose improvements from a legal perspective. This study is of importance to the legal stability of the zoning system and the FARLU in the Republic of Korea. There are differences in the boundaries and areas of cadastral maps displaying property rights among LURMs. In this context, an alternative is needed to replace the inaccurate serial cadastral maps employed in LURMs. Second, the need to introduce tolerance for the LURM, which is different from a cadastral map, is proven from a legal perspective. A cadastral map refers to a map marking the scope of land property rights, with the location of land, parcel, parcel number, boundary, and scale marked [10–12]. Third, measures that will rapidly promote digital cadastral map use, which is map matched with ownership data besides topography, are proposed to replace the old cadastral map approach used with the LURM.

The study is based on the review of relevant literature to identify the legal foundation for the zoning in advanced countries, administrative measures, characteristics of the maps indicating the data and precedent studies. The contents can help readers to better understand the history and development process of the FARLU and LURM. Furthermore, through a case study and the analysis of Supreme Court precedents, the causes of zoning cancellation are clarified before improvement measures are proposed, which justify the proposals from a legal perspective.

Through these improvements, the maps will be able to easily display legal content, thus ensuring that people can conveniently access information. With the development of information technology, the use of maps will be expanded into zoning as well. The authors hope this will support a commitment to ensure the legal stability of zoning and the FARLU and promote research on legal mapping to help legislative and administrative policymakers take a greater interest in using the maps.

The remainder of this paper is organized as follows: Section 2, which provides a literature review, presents the data on document investigations and precedent studies. Section 3 includes an explanation of the scope and methodology of this study. Section 4 identifies the reasons why zoning is canceled and analyzes the results. Section 5 proposes improvement measures, which can be proven to minimize cancellations from a legal perspective. Finally, Section 6 offers some conclusions and suggests future research directions.

## 2. Literature Review

### 2.1. Property Rights, Zoning from an International Perspective, and Maps

2.1.1. Property Rights—International Context

Land is a limited resource, and individual land property rights may be restricted in the public interest [2,13,14]. However, the process of restricting the land property rights of individuals must be fair and transparent, with easily comprehensible due process procedures [15–17].

Zoning, which is a method of restricting land property rights, is the act of establishing a scope for public betterment to enhance the efficiency of or restrict reckless land usage [18–20]. Herein, land usage rights refer to the right of usage and beneficiary rights without disposition rights [21,22].

Zoning restricts personal property rights for public interest purposes, so laws and transparent procedures are crucial. Table 1 presents applicable zoning laws, administrative measurement methods, and guidance used within five major countries. These nations are ruled democratically and have a market economy similar to that of Korea. In the case of the United States (US), court cases have been accumulated, interpreted, and applied in the form of case law. However, as the constitution protects the freedom of individuals and ownership of private property, the laws must be applied according to the constitution [23].

In the United Kingdom (UK), common law and case law are applied instead of constitutional law; thus, the determination and control of zoning is the responsibility of the organization and officials in charge. However, this is supplemented by parliamentary legislation as needed, and the parliament makes complementary decisions through legislation according to each case. In Korea, France, Germany, and Japan, the respective laws are enacted according to codified constitutions, and zoning is performed [23].

From the perspective of administrative measures in zoning, the researched nations collected the opinion of residents, conducted public inspections, and notified residents [23]. In this case, administrative measures refer to the exercising of government authority by institutions to restrict the ownership of private property by its people as a way to enforce zoning [24,25].

In Korea, England, Germany, and Japan, public institutions entrusted with governmental authority or clerical work related to administrative measures are given evaluation and approval rights. However, additional procedures to collect resident opinions and public inspection procedures are in place to prevent infringement on the land property rights of individuals [26,27]. In other words, the above nations are restricting the land use of individuals in regard to zoning designation for the purpose of public good, with the government (administrative branch) as the center. However, concerns that the government might excessively infringe on the property rights of individuals arose; therefore, additional processes to ensure fairness and transparency were established.

Contrarily, in the US and France, the city council holds evaluation and approval rights when it comes to administrative measures of zoning, thereby ensuring the prevention of excessive infringement on the property rights of landowners [28,29]. Restrictions are made by the National Assembly, a body elected by the people rather than the government (administrative branch), which holds land use regulatory power. This was evaluated here as a reinforcement measure to protect individuals' property rights. Moreover, the researched nations all utilized topographic and cadastral maps to mark the zoning [23].

The LURM in Korea and the commune map in France ultimately have been used to assert that zoning has legal power, and thus, the two maps are legally enforceable and have power against third parties. Here, the legal effect refers to the right to restrict the

property rights of individuals and to regulate land use for a zoning designation that will benefit the public good according to respective laws when a zoning map is issued [23–25]. A significant difference in this case is that the maps in the US, England, Germany, and Japan do not have legal effects, but are rather utilized as reports or reference material for the zoning administration [27–37].

2.1.2. Maps—Evolvement and Current Situation in Republic of Korea

Visualization technologies utilizing computers are evolving rapidly [38–40]. Furthermore, maps are facilitating a simpler and more convenient understanding of complex legal principles and legal arguments [41,42].

If you view land use regulation maps such as zoning maps and the laws and related research, as Balchin et al. [1] pointed out with examples from England, it is often obvious that there is a need to narrow the discrepancies in the legal interpretation of maps due to differences in measurement (scale and precision) and differences with respect to previous maps and the development of land use maps (zoning maps) [1].

Andrews et al. [2] stated that starting with the "In re Furman Street" case of Brooklyn's official map in the US in 1836, the US could add restrictions despite the 5th Amendment of Property Protection Regulation of the Federal Constitution. The court also pointed out that "official maps are required to be precise, accurate and legally binding". Andrews [2] mentioned that the official map's legitimacy was in protecting land owner's rights and the balance of public interest. The researchers asserted that the official map would secure the trust of the government's public services, and the government would take infringement of personal property seriously. At the same time, the number of lawsuits could be reduced. Simultaneously, he has pointed out the problems of official maps. According to unforeseen growth changes, the biggest problem is in regard to the uncertainty (inaccuracy) of the official map. For this matter, Andrews suggested that the National Assembly propose specific official map standards in legislation [2].

However, he did not explain what standards should be specifically applied. Beyond the engineering and technical alternatives that were mainly discussed as solutions, it is, however, thought provoking in terms of necessary legislative aspects. This paper also specifically suggests legislative alternatives as a solution to the LURM's problems. Additionally, unforeseen growth changes were pointed out as an issue in the earlier research; this issue is also is a key problem in Korea.

The legal nature, issues, and legal force of France's commune maps are well presented in court cases. Hence, this study examined the latest judicial precedents. According to French judicial precedents [3], after public announcement of the commune map in place of the legal force of urban planning (including land use), they judged that the administrative act did not affect the commune map even if the related administrative act was canceled due to some illegal acts [3]. In addition, according to other judicial precedents [4], the court ruled that the commune map had the same effect regarding whether the legal force of the commune map that took the place of effects of urban planning (including land use) would be valid for reasons such as the enactment of new laws ('ELAN Law Enactment, 9 January 2019) after the public commencement. The ELAN Act is a law on housing, city maintenance, and digital informatization that aims to facilitate the construction of new infrastructure through housing loan regulations and digital informatization [4].

According to the precedents of the French commune map specified above, the commune map was found to be used as data in place of the effects of urban planning and land use and has legal effects that can be acted on the behalf of a third party. It is very similar in legal terms to the LURM in Korea. However, research on these legal and institutional aspects is still very poor.

In other avenues of research, many studies are actively being conducted to solve the problems caused by the integration of these different data types, e.g., by applying engineering and technology solutions to measurement difficulties, cartology methods, matching by using dots and linear objects, and reductions in the discordance of maps through various conflation methods.

Table 1. Comparison of laws, dispositions, and map categories for zoning. Source: National Assembly of Korea (recompiled) [33–37].

| Category | Korea | US | France | UK | Germany | Japan |
|---|---|---|---|---|---|---|
| Applicable laws | • Basic procedures are specified in the FARLU<br>• Specified in 94 individual laws including the Road Act and Housing Act | • Standard State Zoning Enabling Act: SZEA<br>• Standard City Planning Enabling Act: SCPEA | • Land Orientation Law<br>• Law on Solidarity and Urban Renewal | • Planning and compensation law<br>• Planning and forced accommodation law<br>• Housing and urban planning law | • Regional Planning Act<br>• State planning a spatial planning law | • Land Use Planning Law<br>• Urban Planning Law |
| Disposition | • Request of public concessionaire<br>• Public hearing and one-month public inspection<br>• Approval of mayor and governor<br>• Effective after notification of topographic drawings | • City council<br>• Board of supervisors, County commissioners, Trustees | • Hearing<br>• Parliamentary review and approval by governor<br>• Map of commune legally has an opposing power against the 3rd party | • Determined by public organization and responsible officials | • Public hearing<br>• One-month public inspection | • Public hearing<br>• Effective from the date of urban planning notification |
| Type of map | • Produced by superimposing the topographic map and cadastral map<br>• 1/5000 basic drawing<br>• Land use scheme drawing and various drawings are available | • Official mapping<br>• Land use planning map<br>• Thematic maps<br>• Topographic map and cadastral map | • Map of commune (carte communale)<br>• Topographic map and cadastral map | • Os master map<br>• Design guide that describes the level of architectural design<br>• Topographic map and cadastral map | • Zoning map<br>• 1:10,000 design drawing<br>• Topographic map and cadastral map | • No rules. However, information on urban planning, the location of urban planning and progress status is provided in the form of drawings<br>• Topographic map and cadastral map |

Doytsher et al. [8] utilized conflation to solve the problem of using differing datasets of the US Geological Survey (USGS) after the 1980s, and they applied a linear-based conflation complimented on a point-based map [8]. Huh et al. [43] was able to secure reliability and accuracy by suggesting methods for minimizing boundary errors via the application of confidence regions for line segments and points to overlap topographic maps and the cadastral map [43]. Kao et al. [44] highlighted the existence of several problems due to differing boundaries in a cadastral map in Taiwan, which was drawn by using different coordinates; they attempted "the coordinate vector correction method" in conflation to enable concurring conversions [44].

However, zoning has become confusing because of the differences between the LURM and maps drawn in the past, more specifically cadastral maps such as serial cadastral maps. Several researchers are conducting studies on this issue [19,44–46].

Hong [45] insisted that the cancellation of zoning arises from serial cadastral maps not being up-to-date in the LURM. Therefore, Hong argued for a system to update cadastral maps in accordance with cadastral surveying [45,46]; "Serial cadastral map" is a Korean legal term; after digitally scanning each paper cadastral map produced in the past with different scales, the scanned images are attached to one another regardless of the scales. "Serial" in this context refers to a drawing that has several cadastral maps attached in a paper format [12,46].

Moon et al. [20] analyzed the causes of zoning cancellations in Korea between 2007 and 2015. At that time, there were 3316 zoning cancellations. It was found that changes to the plan (2544; 75%) were the largest factor, whereas the second largest was LURM discordance (554; 16%) [20]. Here, changes to the plan refer to changes to the social conditions, including an increase in the population, traffic, and land price, which are spontaneous phenomena [47]. Meanwhile, to prevent zoning cancellations attributable to LURM discordance, conflation of the topographic map and serial cadastral map used for the LURM has been proposed [18,44]. More specifically, affine transformation and rubber sheeting were presented and, in the process, the need for tolerance of the LURM was mentioned [20,48].

The Korea Land Geospatial Information Corporation [49] highlighted that discordances in the LURM could arise from the serial cadastral maps used in the LURM mapping not conforming to cadastral maps. Therefore, as possible improvements, regularly updating serial cadastral maps and linking them with cadastral computerized data systems were proposed [49]. The phrase cadastral computerized data refers to the computerization of data registered to cadastral records, such as cadastral maps. When requests to provide cadastral computerized data (open) are made by the civil petitioner, the case is reviewed and the relevant administrative agency provides a determination. The law defining the responsibility for cadastral computerized data production, management, and organization is titled The Establishment, Management of Spatial Data [50].

As mentioned above, researchers have conducted various studies on the measurement, conflation, system renewal, and map reformation process to recognize and minimize problems related to LURM discordance. However, there has not been any research on the reasons for zoning cancellations due to LURM discordance or on institutional improvements based on legal principles to minimize or prevent it. Such research is critical for maintaining the stability and permanence of laws related to zoning.

One possible reason may be that cartography technology tends to develop more rapidly than the law [51–53]. In contrast, the law tends to remain stable rather than promote changes [54,55]. Therefore, when recently developed cartography technologies, such as the LURM, are used to confirm the effects of the law, the legal effects will be impacted owing to differences between the current and past maps. Hence, research from legal perspectives is needed in accordance with technological and engineering research.

*2.2. FARLU and LURM*

2.2.1. FARLU

Individual property rights can be restricted, if needed, for the purpose of public welfare and efficient use and development of the Korean territory according to Articles 23 and 122 of the Constitution, amended in 1987, and rightful reimbursements must be made [5]. Therefore, the National Assembly enacted FARLU to prevent reckless restrictions of property rights by the government and secure transparency in the protection of individual land property rights and the administration of land usage restrictions [5]. In addition, the law stipulates that restricted individuals must be duly reimbursed [20,22].

The term "ownership right" refers to rights such as usage rights, beneficiary rights, and disposition rights, and "property rights" refer to all economic values, including ownership rights and usage rights [20,22].

FARLU proposes certain basic procedures that must be adhered to during zoning in the interest of public welfare, such as when restricting the use of private property and the specific methods to be used [5,6]. Reviewing this more specifically, zoning-related FARLU serves as the fundamental law. While complying with FARLU, the zoning of 323 locations is determined for public interest according to 128 individual laws. The boundary and area of zoning shall be registered with LURIS for announcement after their indication in the LURM. After such procedures are completed, the legal effect of zoning is finalized to restrict the land use right of the individual [56–59]. This is intended to notify the persons directly involved, and all others, of the fact that the individual land use right is under restriction.

2.2.2. LURM

The LURM is a developed version of urban planning on a cadastral map (UPOCM), an earlier model [23]. The UPOCM refers to a map used to designate and notify people of the scope of land use restrictions in terms of zoning to secure the property rights of the people and transparency in the regulation of land usage under the renewal of the Urban Planning Law in 1971 [20,23].

The UPOCM was maintained even after the abolition of the Urban Planning Law in 2002, through its integration and revision into the National Land Planning and Utilization Act. The UPOCM uses only cadastral maps with parcel boundaries and lot numbers, as shown in Figure 1 [60,61].

Moreover, because cadastral maps in Korea are inaccurate, landowners are unable to determine the exact scope of zoning [61–63]. The cadastral maps are inaccurate because they were created during the Land Survey Project (1910–1918) during the Japanese colonial era (1910–1945) for the purpose of land seizure; the coordinate systems, scales, control points, and types of the land registration system differed for each region [64,65]. The annual litigation expenses have amounted to 38 billion won (USD $320 million) owing to differences in the actual topography and disputes concerning boundaries [66].

However, the government continues to use the current cadastral maps for the permanence of law and safety because these cadastral maps have been used for land property right registration [56,57]. Therefore, the National Assembly and the government enacted the Special Act on Cadastral Resurvey in 2012 to secure legal stability and perpetuity, and remove inaccuracies from the cadastral maps, by which a new set of cadastral maps was prepared identical to the LURM [20,23,66]. They plan to discard the old paper cadastral maps and produce and use digital cadastral maps, on which ownership and actual topography coincide, through the cadastral resurvey project, Cadastral Reform [66,67].

Topographic maps serve as the basic maps in the LURM, which is why cadastral maps are conflated with marked zoning [5,20]. Consequently, land owners and stakeholders could compare the boundaries and areas of zoning with the actual topography and planimetry [5,20,23].

Figure 1 depicts the evolution of the UPOCM into the LURM with changes in law. The set of laws includes acts, enforcement decrees, and enforcement regulations. As depicted in the figure, the Urban Planning Law was revised partially in 1971, which led to the

introduction of the UPOCM. As the Urban Planning Law was replaced by the National Land Planning and Utilization Act, the UPOCM was maintained. However, with the establishment of the FARLU, the UPOCM was abolished and the LURM was introduced. Currently, there are no urban planning laws or UPOCMs.

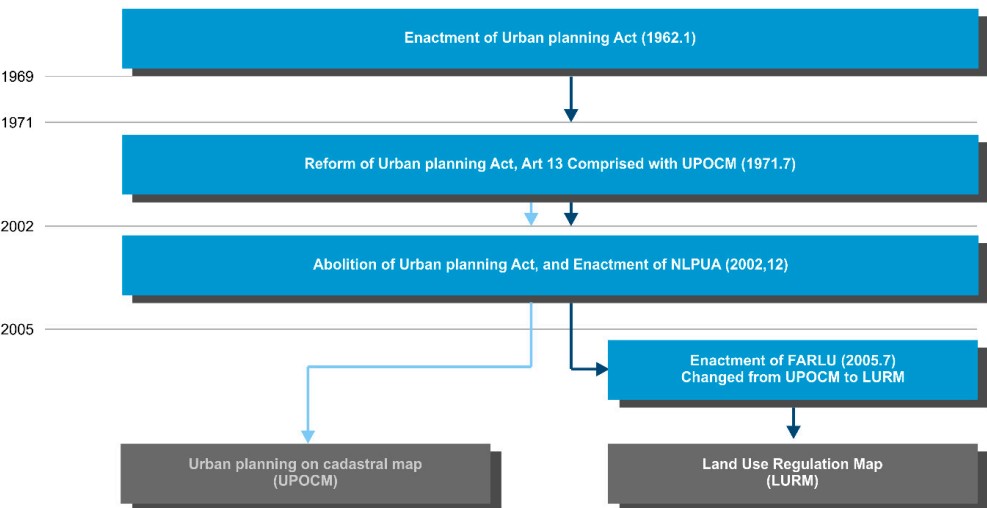

**Figure 1.** Changing legal process and development process from the urban planning on a cadastral map (UPOCM) to the Korean Land Use Regulation Map (LURM).

The differences and advantages of the UPOCM and LURM were studied. Figures 2 and 3 show school zoning with the UPOCM and LURM, respectively, in different regions.

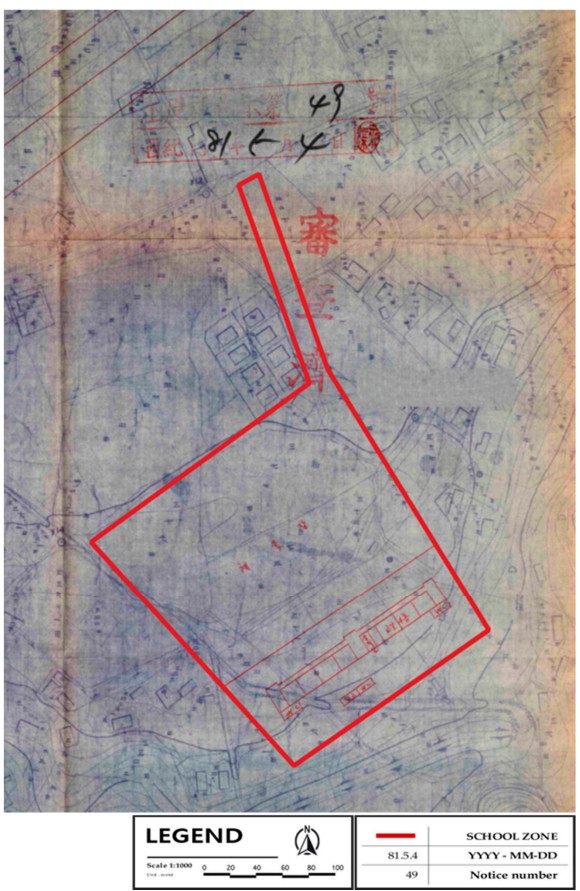

**Figure 2.** Drawing of the UPOCM. Source: Gangwon-do Office of Education [68].

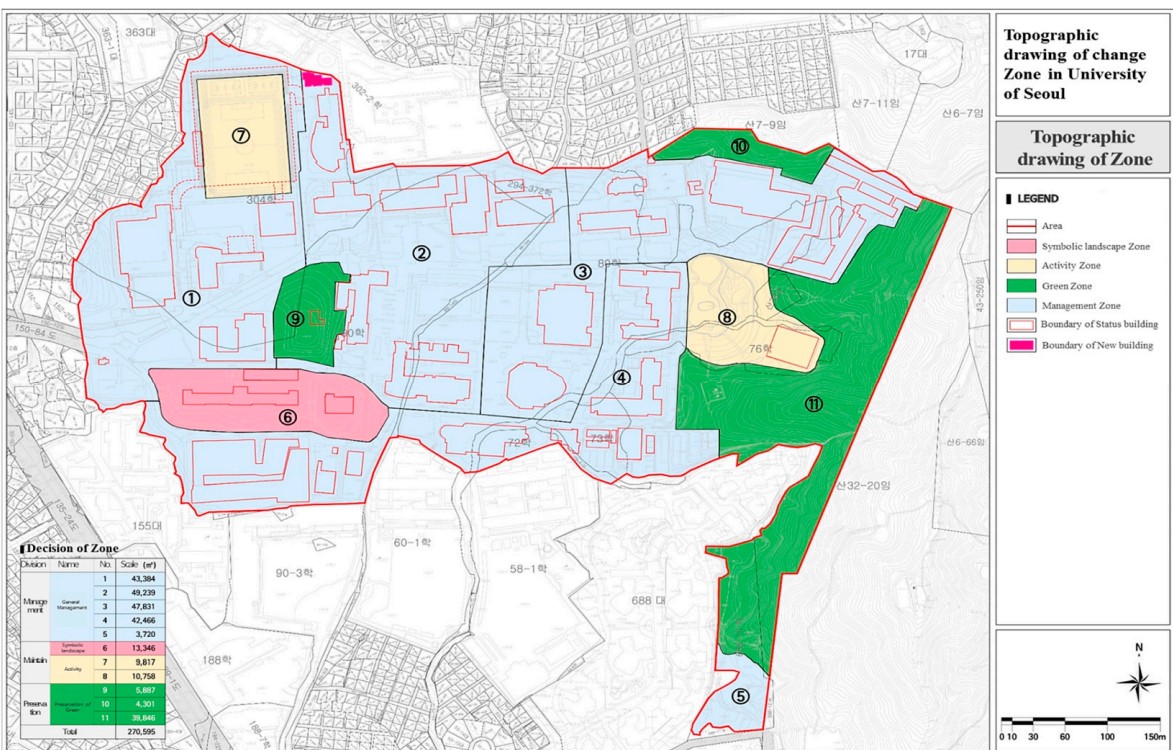

**Figure 3.** Drawing of the LURM. Source: Seoul Metropolitan Government [69].

Figure 2 shows the case of the UPOCM, in which the school zoning (red line) was implemented by the Gangwon Provincial Office of Education in 1981 and designated on the cadastral map. However, the boundary between the school zone and the peripheral region is vague. Moreover, as only a single type of cadastral map was used, it is impossible to compare this map with the topography, i.e., to assess whether there is a mountain or river next to the school. It lacked even basic guidelines, such as direction, scale, drafter, or date. The number 81.5.4 in the middle of Figure 2 indicates the UPOCOM production date (year/month/day), while 49 refers to the notice number.

In contrast, Figure 3 depicts a case of the LURM introduced after the establishment of the FARLU, in which the Seoul Metropolitan Government designated the school zoning in 2017. In comparison with Figure 2, the LURM of Figure 3 enables a clear visualization of the nearby areas and clear boundaries. Additionally, the scale, points of the compass, and areas have been marked, which makes the details of zoning easily comprehensible. There are 128 individual laws for public services, and one zone exists under each law. However, some laws allow 5–10 zones under one law, and each zone contains one LURM. Currently, there are 323 zones; thus, 323 different LURM types exist.

Similar to the UPOCM, the LURM was developed from the perspective of cartography. The changes induced by applying the LURM were researched by examining the number of litigations and Supreme Court cases before and after the introduction of the LURM until stabilization was achieved.

According to Supreme Court precedents, the legal effect of zoning is determined only after the UPOCM has been officially announced in a gazette, and this is applicable even after introducing the LURM. However, technical errors involving boundaries and areas in the production of the LURM shall be identified by a separate review, and the legal effect of zoning shall be determined depending on severity [70–72]. The lawsuit pattern shows that civil actions between individuals were significantly reduced, from 50% (33/66) to 6.5% (1/15) of cases.

In contrast, the proportion of administrative lawsuits against the government or administrative institutions increased greatly from 29% (19 of 66 cases) to 87% (13 of 15 cases).

This is because the entity of notification, designation, management, and responsibility changed to the government or public agencies, i.e., the form of litigation changed from a civil to an administrative litigation where administrative litigation refers to litigations filed against the nation or public institutions [73].

However, the increase in the number of administrative litigations has not been completely negative, as the landowners or stakeholders of zoning can claim their rights from the government or administrative institutions. Given that the purpose of the legislation of the FARLU (LURM) is to protect the property rights of the people and to secure the transparency of zoning, such changes are positive and coherent with the purpose of legislation.

2.2.3. LURM Announcement Process according to the FARLU

If the government seeks to regulate land use for public interest, it can designate 323 zoning districts following 128 acts and subordinate statuses (2020), as prescribed in Figure 4. Regarding the zoning scope, local governments (i.e., public officials) will publicly notify the LURM and register with LURIS so that all citizens have on-line access to it in real-time. Not only that, but anyone can also access LURIS and download relevant content [74]. LURIS is a Korean government web-based service, accessed by an average of 40,000 people daily and provides servicing information on the location, legal basis, and types of zoning in detail to enable use as base material for real-estate trades, urban planning, and policy formulation by public agencies [75].

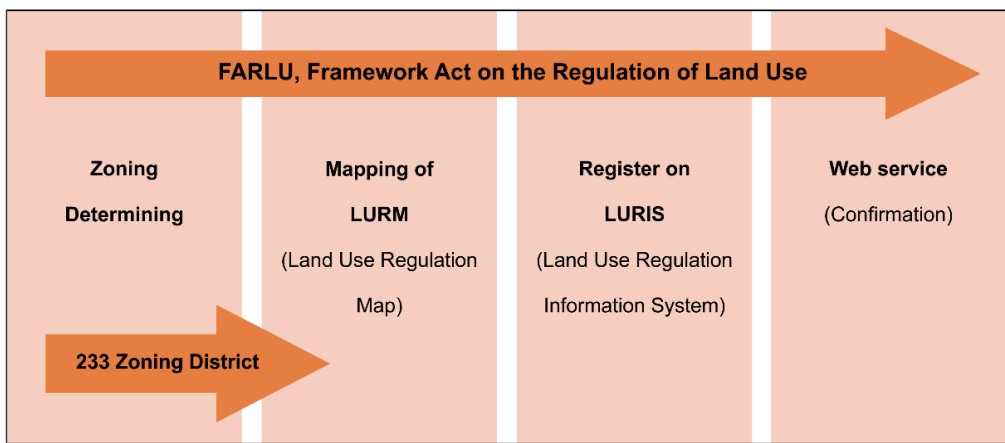

**Figure 4.** Notification process of the land use regulation map (LURM). Source: Ministry of Land Infrastructure and Transport [74].

The Urban Policy Department at the Ministry of Land, Infrastructure, and Transport is responsible for managing the FARLU (LURM). Moreover, the local government officials of the zoning region are responsible for quality assurance and notification in adherence to the guidelines (2012) according to Articles 7 and 8 of the FARLU. Further, the project operator is responsible for preparing the LURM [76–78].

The base map of the LURM is a topographic map in which the coordinate system adheres to the zoning used in the cadastral map [17,56]. With a 1/1000–1/5000 topographic map as the base map (Ⓐ), the serial cadastral map (Ⓑ) is conflated with the zoning (Ⓒ) marked on the map. This results in the final LURM (Ⓐ→Ⓑ→Ⓒ) depicted in Figure 3 [20,73,74].

## 3. Research Scope and Methodology

We attempted to identify ways to solve the zoning map problems in terms of legal and institutional changes, but no relevant papers or reports were found. However, the zoning map's legal problems and legal effects were revealed through Balchin et al. [1], Andrews et al. [2], and the French court's precedents [3,4]. However, it is worthwhile to note that the applicable laws and responsible organizations in regard to the zoning

maps are different, such as those in the US (official map), UK (Os master map), and other countries besides France (commune map) as shown in Table 1.

This paper has a limitation in that its research scope had to be confined to proposals of improvement plans within the scope of the Korean legal system and regulations. Thus, it is not advisable to apply the improvement plans proposed in this paper to each country on the same standard. Nevertheless, the authors argue that this case study on the Korean LURM has significant relevance to other countries, mainly because each country uses topographic maps and cadastral maps prior to the creation of zoning maps as shown in the type of maps in Table 1 above. After all, there is only a matter of a degree of difference in each country; an inevitable difference is generated between cadastral maps and topographical maps. This is because cadastral maps were produced a long time ago compared to topographic maps. In contrast, topographic maps are applied with changes due to urban growth and the latest cartography technology. Ideally, the improvement plans proposed in this study may be used as a conceptual framework or reference material according to the circumstances of each country in the future regarding the legal cancellation of the zoning caused by discordance between topographic maps and cadastral maps, as exemplified by the problems associated with the Korean LURM.

This paper's research methodology involved collecting cases in which zoning was canceled due to the discordance of the LURM, which is Korea's zoning map, to identify the causes of why this occurs. Additionally, precedents of the Supreme Court of Korea were analyzed to determine what disputes might have arisen due to zoning cancellation. The results from the above analysis showed that the following factors were problematic: ① incorrect data were used for the LURM, and ② incorrect allowable errors were applied to the LURM. Therefore, ③ this paper proposes improvement plans based on the findings. At this time, it can be proven that these proposals are correct from a legal point of view.

The proposals are being suggested for legislation by the lead author of this research as a legislator. Furthermore, similar proposals have undergone deliberations by the National Assembly of Korea and have been allowed to proceed (31 October 2019).

## 4. Causes of Zoning Cancellation Identified through LURM Cases and the Literature

The purpose of this study was to identify the causes of zoning cancellations in the enforcement of zoning maps through the case of the LURM and suggest possible enhancements from a legal perspective. Therefore, prior domestic and international research, the history of laws related to the LURM, preparation and notification processes, and legal effects were explored during the literature review. We identified the causes of zoning cancellation through local examples.

*4.1. Discordance of the LURM May Represent an Infringement on Property Rights. Why Is the Incorrect Serial Cadastral Map Still Being Used When Cadastral Computerized Data Are Available?*

4.1.1. Infringement on Property Rights According to Case Studies

Legal disputes on zoning have been significantly reduced since the LURM was introduced. However, as revealed in the study of Moon et al. [20], the zoning designated for the purpose of public goods after the LURM enactment in 2005 is being cancelled [15,20]. Why did this occur? The authors clarify the cause through a case study on Jeongeup-si (692.7 km$^2$, population 130,000). Jeongeup-si was adopted because it had less changes to the city plan and is a flat area of land where urban and rural areas coexist.

Figure 5 shows Street Nos. 667 and 667-1. Sangam-dong, Jeongeup-si. Six zones (small river, national park, natural environment, restriction on livestock, production management, and environmental conservation) were designated for Street Nos. 667 and 667-1, which are targeted urban planning regions for reasonable land use. Jeongeup-si City notified the public that six zones were designated along Street Nos. 667 and 667-1. The six zones are not shown on the LURM, but are provided in text format as land usage restriction details in the LURIS.

However, as a result of comparisons with (a) by using the cadastral map (blue lines) consistent with the ownership as shown in (b), only Street No. 667-1, excluding Street No. 667, fell under the region of the land use regulations. This resulted in the three zones (small river, restriction of livestock, and production management) for (b) instead of the six initial ones. Additionally, even though three zones in (b) were canceled, some differences were found in the details of the land use regulations such as in (a) initially.

Why, then, was zoning cancelled? It was not cancelled because of a technical issue, but a legal one. As stated above, zoning is defined in a map indicating the restriction of the property use right, which is a property right of the people. When there is discordance with the cadastral map, indicating the range of ownership in terms of the boundary and area, the relationship of rights and duties becomes vague, and the zoning is inevitably canceled until such relations are clarified [61,62,70,71].

In addition, until a public project can be continued by changing the LURM (a) from a public project to a cadastral map (b), the landowner of the public project is prohibited from exercising property rights, including disposal, removal, and reconstruction and extension during the months of administrative changes, including the collection of resident opinions and the holding of a deliberative body [25]. This is against the principles of statutory reservation ("Vorbehalt des Gesetzes") [79–81], which dictates that details of people's rights and obligations must be regulated by the law. In other words, the details regarding property rights are the fundamental rights of the people, which cannot be restricted without related laws. Thus, while individual property rights could be restricted for the purpose of public welfare, details of restrictions must be codified in respective laws and cannot be applied when there are no relevant articles in the law [81].

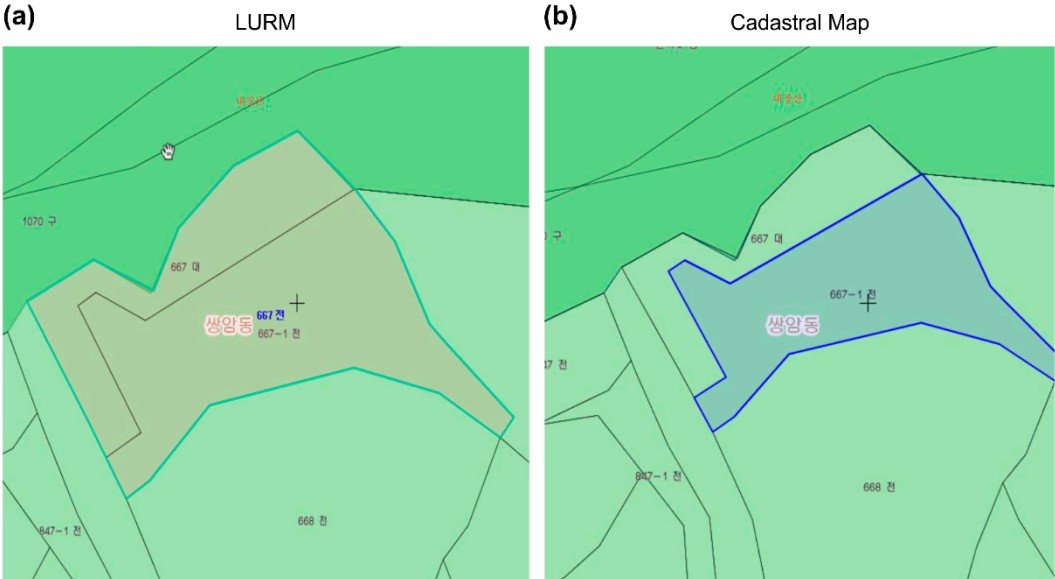

**Figure 5.** Discordance between the LURM and cadastral map. (a) Public project area (brown area); six zones (small river, national park, natural environment, restriction of livestock, production management, and environmental conservation). (b) Public project area (blue area); three zones (small river, restriction of livestock, production management). Source: Chonbuk Provincial Government [82].

4.1.2. Incorrect Data (Serial Cadastral Map) Is Applied to the LURM

As the LURM shows the scope of restrictions of usage rights, it is critical that it is updated and ensured for notification in accordance with relevant property rights [20,49]. Moreover, given that the data are being used in the real-time service LURIS, served by the government to the people, and since more than 90% of the data are being used in relation to property rights, including sales of buildings and land, confirmation of land use plans, and urban planning, current data must be ensured [20,47].

However, the government has dictated that it is compulsory to use serial cadastral maps in preparing the LURM in an enforcement degree, a decree for carrying out the FARLU [5]. Therefore, this is one of the causes of zoning cancellation. The government is already aware of the difference between serial cadastral maps and cadastral maps at the same location. LX, the government agency responsible for the cadastral survey, twice revised and supplemented the serial cadastral maps on as many as 101,808 sheets of cadastral maps in 2008 (1st) and 2009 (2nd) [47].

According to studies by LX, which compared serial cadastral maps and cadastral maps, in 1013 cases (93.36%), the land area differed by 0–1%; in 36 cases (3.32%), the land area differed by 1–10%; in 6 cases (0.55%), the land area differed by 10–20%; and in 12 cases (1.11%), the land area differed by 20–100%. In 16 cases (1.47%), the cause was unidentified [24,47].

Why does the concerned authority continue urging the use of serial cadastral maps for the LURM? In this regard, the following facts were revealed. First, the office of concern considers the LURM as the reference or document for zoning, as in the previously reviewed foreign case and as in the US, Germany, and Japan [26,27]. However, even among the Ministry of Land, Infrastructure, and transport-affiliated organizations, departments related to spatial information and the production of and management of maps claimed that using serial cadastral maps in the LURM restricts property rights, that this practice is causing discordance with the cadastral map, and that processes must be improved, as this will lead to the infringement of property rights [23,26,27].

The judgment on who is right lies in the law and ruling cases of the Supreme Court. Therefore, the judgment of the office in charge of land use regulations could be identified as erroneous in terms of interpreting the law arising from a misunderstanding of the relationship of usage rights and property rights from legal perspectives, and a lack of understanding of laws, including the FARLU and The Establishment, Management, of Spatial Data, and the characteristics of the LURM. As shown in a Supreme Court case and prior research, the LURM has been upheld as a means of confirming the zoning [61,62,70,71].

Second, according to the National Assembly Research Service [6], the office concerned urged the use of serial cadastral maps in the LURM because the maximum distance that can be indicated on a sheet of a cadastral map is approximately 300–400 m. Thus, use of a cadastral map to produce an LURM for a railroad, road, or housing development site that usually extends to several kilometers is inconvenient, time-consuming, and costly. As serial cadastral maps have already been structured nationwide, using them is convenient, and thus, these are enforced [6,20,45,46].

As shown above, due to errors in the interpretation of the law, a lack of understanding of Supreme Court cases, and the comparative convenience of serial cadastral maps for large-scale social overhead capital (SOC) projects, improper maps for the LURM are being applied.

### 4.1.3. Use of Cadastral Computerized Data, Including Cadastral Maps and Digital Cadastral Paps, Is Restricted

As indicated in Section 4.1.1, the discordance of the LURM differs from the scope of ownership and is regarded as an infringement on property rights, according to the Supreme Court precedent, and zoning is canceled accordingly.

Why, then, was the LURM not compared with the cadastral computerized data, including the cadastral map coinciding with the ownership and the digital cadastral map prior to announcing the LURM? The reason is that providing cadastral computerized data, including a cadastral map and digital cadastral map, was selectively allowed owing to deliberation by the office concerned, according to Article 76 of the current The Establishment, Management, of Spatial Data [12,23,83]. In fact, this is closer to non-disclosure than disclosure [20,50,84]. Accordingly, it is impossible to compare the cadastral computerized data, including the digital cadastral map, with the LURM, which is an infringement of the people's right to know in terms of the principle in the constitution. The right to know refers to the right of individuals to obtain information regarding politics, society, and reality as

sovereigns. For instance, let us suppose that an individual applied for issuance of an ID to a government office. When the applicant demands information on when the ID was made and the processes of issuance, the government office must provide that information, which is directly related to the freedom of speech [85,86].

*4.2. Inappropriate Application of the Tolerance of a Cadastral Map to the LURM*

In the process of preparing a map, quality assurance entails evaluating the accuracy of the map and its credibility [87–89].

However, the current guidelines do not include the legal definition of the LURM, specific methods for deciding the coordinate system, conflation methods, and common point selection methods [5,20]. Moreover, before notifying the LURM, there is no allowable error in the LURM that could be provided to the civil servants of local governments for comparisons with the cadastral map [43,44,59]. The common point refers to the coordinates of a location to conflate different maps [60,90]; the margin of error refers to the difference between the set maximum and minimum values [72,91].

Although the allowable error in the LURM is fundamental for quality assurance, offices that are primarily concerned with the execution of the law and policies are currently applying this error in cadastral maps because the LURM is a map that contains information on the restriction of property rights [20,23]. However, this is a contradiction, as the relevant offices assume the LURM to be a mere attachment.

However, the perspectives of the involved offices and local governments and civil servants in the management of LURM notifications differ [5,74,75]. The reason for this is that if there are differences in the notification details from the actual notification, the local government (civil servant) in charge of the notification may be subject to administrative litigation and may have to provide compensation when negligence is proven [20,80]. The term notification refers to administrative measures that may be subject to appeals when used to restrict the rights and obligations or the fundamental rights of people [73]. In other words, since the LURM restricts property rights as a fundamental right of the people, when the truth differs from the notification, the local government and civil servants may be held responsible for the compensation for the litigation.

The current allowable error in a cadastral map is $3/10 \times M$ (mm), and the allowable error in an area is $A = 0.02M$ (A: allowed area, M: scale denominator) [5,90].

For instance, for a map of scale 1/1200, the allowable error in the cadastral map distance would be $\pm 36$ cm. If the scale is 1/1200 and the area is 1 km$^2$, then the allowable area error in the cadastral map would be $\pm 0.63$ km$^2$. However, there arises a question of whether this allowable error is appropriate from a legal perspective. This allowable error stipulates that only cadastral maps should be used for real estate transactions. However, is it appropriate to apply the error tolerance of cadastral maps to LURMs used for land use regulations for public interest purposes, but not to LURMs used for real estate transactions?

Before considering the perspectives of mapping, including surveying, cartography, and computer science, it is improper from the legal perspective to establish and interpret the law.

The reason for this is that the principle of proportionality, one of the general principles of the law, and the legislative power of the legislator that legislated the FARLU (LURM) have been infringed upon. The principle of proportionality is a fundamental principle of the law dictating that a rational proportion should be maintained between the realization of a goal and its means. For instance, when traffic accidents due to drinking occur, versus traffic accidents due to negligence, the driver is subject to additional punishment [79,90]. Further, the legislative power, as one of the powers guaranteed by Article 40 of the constitution, guarantees judgment rights, and the formation and temporal details of the legislation. Discretion related to such legislation is widely acknowledged within the scope of the constitution [57,79,80].

Therefore, there is a need to introduce allowable error in the LURM, which is separate from that of the cadastral map. However, to introduce such error from a legal perspective,

the distinction between the LURM, i.e., markings of land usage right restrictions, and the cadastral map, i.e., markings of the scope of ownership rights, must be proven.

*4.3. The Digital Cadastral Map Preparation Speed Required for the LURM Is Slow*

According to the Special Act on Cadastral Resurvey, preparation of a cadastral map, including a serial cadastral map, as a digital cadastral map identical to the LURM is recommended [5,6]. However, according to the government's data submitted to the National Assembly, the target progress from 2013 to 2017 was 16% of the total, but the actual progress was very low, only 7.2% [47]. This delay was attributable to a budgeting problem and the time required for the Cadastral Reform Administration [23]. To reduce the time required for administrative processes, the authors propose the introduction of full-scale legal fiction to 22 laws for The Establishment, Management, of Spatial Data for Cadastral Reform efforts. Legal fiction refers to acknowledging certain contents in other laws applied for public interest for institutional purposes, with legal statements such as "it is assumed to be" or "it is considered to be" [5,50,63]. For instance, if the legal fiction of item Ⓐ is reflected in law A, related laws of B, C, D, and so on, item Ⓐ would be acknowledged without revision of the respective laws.

Legal fictions are used widely in legal systems in Korea, with 116 laws being applied, including 32 concerning the construction sector, 7 concerning the transport sector, 16 concerning the energy sector, and 54 others that do not correspond to any sector [50].

## 5. Proposals from a Legal Perspective Needed for LURM Enhancement to Minimize Zoning Cancellation

A cause of cancelled zoning in the aforementioned LURM cases was that erroneous data (i.e., the serial cadastral map) were used, and the requested data (i.e., cadastral computerized data) were restricted by the government. Moreover, an allowable error was being applied erroneously to the LURM, and the preparation speed of the digital cadastral map (i.e., data required for the LURM) was slower compared with the plans. Therefore, legal enhancements are needed to resolve these issues.

*5.1. Make the Cadastral Computerized Data Public and Apply It to the LURM While Stopping the Use of Serial Cadastral Maps*

This paper proposed not only the renewal of serial cadastral maps or the connection to the cadastral computerized data system to improve the LURM, it also suggests that the use of serial cadastral maps should be suspended and the cadastral computerized data made public on a full scale to apply it to the LURM. As explained previously, the cadastral computerized data are what are computerized into the database from a cadastral map, which coincides with the ownership registered at the competent court.

By producing the LURM, making the cadastral computerized data public (as in Figure 5), and using the cadastral map (b) instead of the incorrect serial cadastral map, the discordance with ownership could be mitigated or eliminated because the cadastral map matching with the ownership is incorporated previously at the stage of production of the LURM.

In addition, as the LURM indicates the range of usage right restrictions, it relates to property rights, which are one of the fundamental rights. Despite that, the review and restriction of the cadastral computerized data, which are critical data relating to property rights, by the office concerned must be viewed as an act of arrogation, and against the Principle of Statutory Reservation, which shall be corrected.

In addition to the previously stated problems, discordances in the LURM occur, as shown in Figure 5, because the responsible offices direct the use of inaccurate serial cadastral maps in the LURM, thereby resulting in the cancellation of the LURM. As indicated through the court case analysis, the Supreme Court is acknowledging the LURM as a means of confirming the legal effect of zoning.

However, the relevant offices maintain the application of the inaccurate serial cadastral maps in the LURM, thus, seeming to oppose the judgment of the judicial body according to which improvements must be implemented. Furthermore, the application of inaccurate

serial cadastral maps in the LURM causes zoning cancellation that does not qualify as a rational reason because the application of a single-sheet cadastral map to SOC is complicated.

Therefore, the following legislative enhancements are proposed: permission of using the cadastral computerized data with the LURM in FARLU Article 8 (Designation of Zone, District, and so on), Enforcement Decree of the same law, and Article 7 (Method of Preparation and Public Announcement of Topographic Drawing), and abolishing the judgment of administrative agencies related to The Establishment, Management, of Spatial Data Article 76 (use of cadastral computerized data, etc.) to allow the disclosure of the cadastral computerized data and use of the cadastral computerized data in the LURM rather than serial cadastral maps to prepare the LURM.

### 5.2. Demonstrate the Justification for Introducing the Tolerance of the LURM from a Legal Perspective

The LURM indicates the scope of restricting land usage rights, which is one of the three rights included in property rights. However, property rights refer to strong disposable rights—usage and beneficiary rights—that differ in terms of legal characteristics. Additionally, it infringes upon the principle of proportionality applied for the application and interpretation of the law [81–84]. Therefore, an alleviated allowable error standard for the LURM must be applied instead of the allowable error of land registration rights.

Furthermore, applying the same principle of cadastral maps to the LURM is a severe infringement of the National Assembly's legislative power legislating the FARLU and the authority of the designator designating zoning [57,79,80]. The legal basis of the LURM and cadastral map are the FARLU and The Establishment, Management, of Spatial Data, respectively, with different legislative purposes.

Hence, to respect the legislative power of the National Assembly legislating the FARLU (LURM) and to achieve legislative goals, the cancellation of the zoning designated for public interest should be minimized by introducing an allowable error in the LURM to ensure the legal stability of the FARLU.

The following improvements are proposed: when altering zoning due to differences of the boundary and area between the cadastral map and LURM, the legal effect of zoning must be maintained when the error of the LURM is within the allowable error. If the error of the LURM exceeds the allowable error, the infringement of property rights must be acknowledged, and the LURM must be prepared again along with notices.

However, the 128 respective laws regulated by the FARLU have different legislative purposes and subjects of application. Therefore, rather than proposing a single allowable error for the FARLU, it is preferable to propose a set of allowable errors based on respective laws, through future studies.

### 5.3. Introduce Legal Fiction for the Digital Cadastral Map to Achieve Timelier Preparation of the Digital Cadastral Map Needed for the LURM (Fast Track)

As mentioned previously, use of old incorrect cadastral maps, including serial cadastral maps, for producing the LURM is one of the major causes of the cancellation of zoning. Thus, the National Assembly and the government plan to produce digital cadastral maps that coincide with the ownership and topography through the Special Act on Cadastral Resurvey by 2030 to eliminate this conflict. Should the digital cadastral maps be applied to the LURM, the digital cadastral map that coincides with the ownership will be incorporated into the LURM, thereby minimizing the cancellation of zoning due to the discordance of the LURM. However, the production of digital cadastral maps has been delayed.

Therefore, this paper proposed the introduction of legal fiction to The Establishment, Management, of Spatial Data, which is the base law for digital cadastral map preparation, to expedite the administrative procedure of cadastral reform, which is one of the causes of delays in the preparation of digital cadastral maps. This is because digital cadastral map preparation requires that separate approvals must be received for each of the 22 laws of The Establishment, Management, of Spatial Data.

However, if approval-related legal fiction for cadastral maps is applied to The Establishment, Management, of Spatial Data, because approval was already made in the law, the 22 laws and lower statutes do not need additional approval processes. Thus, an effect similar to a fast track would be achieved.

Therefore, as a legislator, the lead author of this study inquired as to the opinions of relevant government agencies regarding seven laws related to digital cadastral map preparation obligations as shown in Table 2. As a result, he received a reply stating that legal fiction on five laws (Housing Site Development Promotion Act, Sports Facility Act, Tourism Complex Act, Road Act, and Railway Act) was needed (19 July 2019). However, the Ministry of Maritime Affairs and Fisheries (Public Waters Reclamation Law) rejected the inquiry as an additional review of related maritime laws was required because the subject of the law was the sea, and not the land. However, the ministry acknowledged the necessity of legal fiction. In addition, the Ministry of Land, Infrastructure, and Transport (Housing Act) stated that opinions on whether the laws will be finally accepted after being partially accepted would be required. This is because housing is an integral part of people's lives, which makes the subject and content of the inquiry expansive, and because a review on the possibility of a cost change is needed [5,20,23].

**Table 2.** Results of government opinion inquiry on law revision to make legal fiction mandatory in digital cadastral map preparation. Source: Ministry of Land Infrastructure and Transport [23].

| Zoning | Revised Law | Relevant Government Agency | Acceptance Status |
| --- | --- | --- | --- |
| Housing Development Area | Housing Act | Ministry of Land, Infrastructure, and Transport | Partially Accepted |
| Housing Site Development Area | Housing Site Development Promotion Act | Ministry of Land, Infrastructure, and Transport | Accepted |
| Sports Facilities Area | Sports Facilities Construction Act | Ministry of Culture, Sports, and Tourism | Accepted |
| Tourism Zone Area | Tourism Promotion Act | Ministry of Culture, Sports, and Tourism | Accepted |
| Reclamation Project Area | Public Waters Reclamation Law | Ministry of Maritime Affairs and Fisheries | Rejected |
| Railroad Area | Railroad Act | Ministry of Land, Infrastructure, and Transport | Accepted |
| Highway and General Road Area | Road Act | Ministry of Land, Infrastructure, and Transport | Accepted |

Thus, most stated their will for acceptance since embarkation, changes, and completion must be reported to administrative bodies to prepare a digital cadastral map. However, if legal fiction is applied as a possible improvement and if the implementer has acquired the approval for the primary execution plan, the following digital cadastral map-related project embarkation is assumed to have been completed; hence, the time taken for administrative procedures could be shortened.

The introduction of legal fiction in the Land Development Act and the other four cases (a total of five cases) was proposed through legislation by Assembly members (Legislation No. 9983 and other four cases, 2 November 2017) [23,50]. The legislation has been implemented after voting (31 October 2019) and the opinion of the National Assembly's legislation evaluation committee was coherent with the National Assembly Research Service, according to which the time taken in SOC projects and others could be reduced. Therefore, the promotion of the expansion of this law is proposed. Furthermore, it must be noted that this improvement has been promoted by the author while serving as a legislative policy officer at the National Assembly. The proposed possible improvements may be utilized in related research, including those studying cadastral reforms. In the future, the findings are expected to assist countries that set forth on the path of cadastral reform, such as Canada, the US, and Australia, in terms of both legal and regulatory aspects.

## 6. Discussion and Conclusions

This study aimed to surpass the technical aspects of research that minimizes the inconsistency between boundaries and map areas displaying zonings designated for public interest purposes. It identified the reasons owing to which the legal effect of zoning led to cancellation due to inconsistencies with the cadastral map and suggested an alternative. Moreover, this study attempted to enhance the stability and permanence of zonings by suggesting the legalistic theory.

The Korean National Assembly and the government have attempted to contribute to the transparency, simplicity, and explicitness of zoning and restricting the property rights of people through the introduction of the LURM. However, the LURM has a few disadvantages. First, zoning is being canceled due to LURM discordance, thereby leading to a declining stability and trust of the FARLU and related respective laws. Second, inaccurate serial cadastral maps are being applied to the accurate LURM preparation process. Third, there is no allowable error for LURM quality assurance.

Therefore, this paper proposed the full disclosure of data (cadastral computerized data) and the application of these data to the LURM to resolve the issue of discordance. Additionally, in this study, the appropriateness of the introduction of an allowable error in the LURM was verified from a legal perspective. Third, the expansion of related laws to transform an inaccurate cadastral map—the fundamental underlying problem of the LURM—into a digital cadastral map was proposed. The proposal includes fast-tracking to reduce the time required for cadastral and administrative reforms. Among the laws related to the digital cadastral map, five laws have passed voting at the National Assembly (31 October 2019) and are currently being implemented.

However, as mentioned before, there are limitations of applying LURM enhancement to each nation. This is because legal systems related to zoning differ according to the selected bodies of law and economic management of each nation. The background, ideologies, and conflicts at the time of legislation of laws related to zoning must be studied for each nation. This will allow the suggestion of legal enhancements for each nation according to zoning map discrepancy. Therefore, related follow-up research is needed.

Furthermore, follow-up research from technological and engineering aspects is anticipated.

For instance, conducting cost–benefit analyses of applying the cadastral computerized data to the LURM, categorizing the LURM according to the respective law, continuing research of the related actual conditions, proposing allowable errors according to the respective laws, and analyzing the economic feasibility after the application of legal fiction can be some of the directions for future research.

As shown in the LURM, cartography and surveying techniques develop gradually. This will inevitably lead to a greater discordance with the maps produced in the past. Despite such problems, maps are able to specifically present the laws with the development of scientific technology and cartography. Hence, they can be prepared more easily and conveniently, thereby leading to increased use of maps.

Therefore, additional efforts to elucidate improvements from legal perspectives or from legal mapping activities and research regarding discordances in maps, thereby confirming the legal effect in conjunction with the law, or instead of the law, from technical and engineering perspectives, would be considerably beneficial.

**Author Contributions:** Conceptualization, J.-k.M.; methodology, H.-g.S.; formal analysis, S.-b.Y. and Y.-s.C.; supervision, H.-g.S. The authors jointly edited the article. All authors have read and agreed to the published version of the manuscript.

**Funding:** This research received no external funding.

**Data Availability Statement:** Not applicable.

**Acknowledgments:** In the process of preparing this manuscript, data and information were provided by the office of lawmaker Kim JungRo, the National Assembly of Korea, the National Assembly Research Service, the National Assembly Library (Overseas Data Department), and the Engineer Surveying Geo-Spatial Information Center, Yonsei University.

**Conflicts of Interest:** The authors declare no conflict of interest.

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
