# Peer review of "Conflicting Maps: How Legal Perspectives Could Minimize Zoning Cancellation in Republic of Korea"

_land, doi:10.3390/land10030256_

Round 1

Reviewer 1 Report

Journal

Land (ISSN 2073-445X)

Manuscript ID

land-1079174

Type Article

Number of Pages 25

Title

Conflicting maps: how legal perspectives could minimize zoning cancellation in South Korea

It is a correct manuscript and is a scientific contribution to the Land journal from their experience. I make a few minor observations as a minor revision in case the editor and authors would like to take them into account.

  • Writing style. Try to avoid writing in the first person... We....
  • Although there is a recommendation in the abstract. I suggest improving the main recommendation based on the results at the end of the summary by 1-2 sentences.
  • It is preferable to have a separate Discussion section to discuss the results obtained. To enrich the structure of the manuscript and improve readability, I suggest that there should be a discussion section before the conclusions where the main comparisons with previous work are collected.

Reviewer 2 Report

The paper presents a detailed and interesting analysis of the phenomenon, presenting the situation in other countries. It would be important for the conclusions drawn from this material to be implemented in practice

Reviewer 3 Report

The publication presents very interesting possibilities for solving the presented problem. It is possible to implement the proposed solutions in other countries.

Reviewer 4 Report

The paper discusses legal perspectives for spatial planning (zoning plans) in the context of cadastral maps. The topic is interesting and relevant for the journal.  However, the paper requires improvements in several places (mainly better explanations of concepts):

  • I do not unterstand the difference between "cadastral maps" and "serial cadastral maps". Since serial cadastral maps are relevant for the solution, an explanation is essential for the paper.
  • Lines 541 ff: Allowable error is not only necessary to account for deviations in the cadastral maps but also for deviations in reality. Allowances provide practitioners the possibility to avoid excess effort by slightly adapting the planning to reality. It may, for example, be the case if there is an overlap of 1cm between a wall and a planned road. Moving the wall would cause more costs (in terms of money and time) than reasonable for an economically and socially reasonable solution.
  • Legal Fiction: As a surveyor, I have problems understanding the concept. The references are not helpful (in the thesis [63], the term "legal fiction" is not used ...). Please provide a concrete example - it should help to understand the concept.
  • Figure 1: "Law" and "Act" are mixed (Urban planning law in the text, urban planning act in Figure 1). Please harmonize. If UPOCM was abolished with the introduction of FARLU, why is it still in Figure 1 at the bottom left?
  • Figure 2: The map is extremely difficult to reda. Would it be possible to scan it with a higher resolution and use a smaller part of the map?
  • Figure 3: Is there any geographical reference except for "University of Seoul"?
  • Figure 5: I do not see the zonings represented in the figure but they are mentioned in the text.
  • Lines 61/62: "The area of the current LURM is roughly 4.3 times (432,390 km2) the territory of the Republic of Korea (99,720 km2)." Does that mean that there are on average 4.3 different sets of regulations for each location? Why? Could you provide more detail?
  • Lines 157-161: I am not sure if I understand what the authors want to say. Maybe the sentence is too long.

minor issues:

  • Please use term consistently (South Korea vs. Republic of Korea)
  • Lines 51/52/53: "zoning has the legal effect of limiting individuals’ property rights (right of use) only after producing the LURM, a requirement unique to the FARLU, and registration in the official gazette and Land Use Regulation Information System (LURIS) is required" - How could a legal effect start otherwise (without proper documents initiating the effect)?
  • Line 76: What is different? The LURM or the tolerance?
  • Line 410: Are the areas 667 and 667-1 streets or is street No. 667 an address?
  • Line 525: F is not used, is it there by accident or is a formula missing?
  • The links provided with references 5 and 6 do not work

Author Response

This manuscript is a resubmission of an earlier submission. The following is a list of the peer review reports and author responses from that submission.

Round 1

Reviewer 1 Report

The proposed position paper addresses the cancellation of the legal effects of zoning in South Korea. The manuscript is quite hard to read, in that the Authors let the ‘track changes’ in the PDF also for the deleted sentences. Furthermore, this reviewer remarks the weaknesses of the proposed method (Section 3), which appears to be scarcely rooted in scientific literature although the Authors report on three publications. In other words, the Authors did not justify the proposed method, which seems to be based on personal considerations. Finally, the manuscript is strictly tailored to the South Korean context and the Authors did not discuss the relevance of their findings with respect to the international scientific literature. In my opinion, the manuscript is not ready for publication.

Major issues

The Abstract is almost unreadable because of the deleted sentences.

Lines 372-388: the paragraphs are really difficult to read because of the deleted sentences and this reviewer has scarcely understood what the Authors intend to communicate.

Figure 4 lacks in terms of comments in the body of the manuscript.

Reference [32] refers to Wikipedia. This reviewer acknowledges the usefulness of Wikipedia for common people, but feels it is not a proper reference in a manuscript submitted to a prestigious international scientific journal with impact factor.

The method appears to be scarcely rooted in scientific literature although the Authors report on three publications. In other words, the Authors did not justify the proposed method, which seems to be based on personal considerations or perspectives.

Lines 752-763: the paragraphs are really difficult to read because of the deleted sentences.

Reviewer 2 Report

Very nice manuscript, I like originality and novelty. I recommend publication in the present form. 

Reviewer 3 Report

I state that the reading of this article was very complex because the pdf file containing all the changes by the authors was sent.

Therefore, first of all, I would suggest the authors to send their papers without their own revision for a simpler and more linear reading.

I also encountered considerable difficulties in understanding the topic which has a mainly technical content with specific regulatory reflections for the country of the authors but which, for example, from my experience, has no response in my country (Italy).

So the paper is characterized by a very specialized topic valid only in the country of origin of the authors.

Among other things, in fact, even the comparative analysis with five countries (USA, GB, F, J, D) carried out at the beginning does not go into the specific details of the problems highlighted by the authors linked to the use of different cards, but offers just a general overview of planning and zoning.

I would suggest, therefore, if possible, to check if even in these countries (USA, GB, F, J, D) there is a problem linked to the errors deriving from different cartographic bases and how these have been solved.

Specifically, even the excessive use of acronyms (LURM, CM, DCM, CCD, CSM, FARLU, LURIS, NLPUA, SACR, etc.) does not make it easy to understand the paper and, therefore, their use isrecommended only if strictly necessary.

A specific question is on fig. 5 where the LURM and the CM are compared by referring to codes (in the text) but which cannot be read in the figure itself.

Ultimately, I believe that for the publication it is necessary to identify some more general elements that may interest a wider audience of researchers.